# Planar refraction and lensing of highly confined polaritons in anisotropic media

J. Duan[1,2,12], G. Álvarez-Pérez [1,2,12], A. I. F. Tresguerres-Mata [1], J. Taboada-Gutiérrez [1,2], K. V. Voronin [3], A. Bylinkin [4,5], B. Chang[6], S. Xiao [7], S. Liu[8], J. H. Edgar [8], J. I. Martín[1,2], V. S. Volkov [3,9], R. Hillenbrand [10,11], J. Martín-Sánchez [1,2], A. Y. Nikitin [5,10] & P. Alonso-González [1,2✉]

Refraction between isotropic media is characterized by light bending towards the normal to the boundary when passing from a low- to a high-refractive-index medium. However, refraction between anisotropic media is a more exotic phenomenon which remains barely investigated, particularly at the nanoscale. Here, we visualize and comprehensively study the general case of refraction of electromagnetic waves between two strongly anisotropic (hyperbolic) media, and we do it with the use of nanoscale-confined polaritons in a natural medium: α-MoO$_3$. The refracted polaritons exhibit non-intuitive directions of propagation as they traverse planar nanoprisms, enabling to unveil an exotic optical effect: bending-free refraction. Furthermore, we develop an in-plane refractive hyperlens, yielding foci as small as $\lambda_p/6$, being $\lambda_p$ the polariton wavelength ($\lambda_0/50$ compared to the wavelength of free-space light). Our results set the grounds for planar nano-optics in strongly anisotropic media, with potential for effective control of the flow of energy at the nanoscale.

[1] Department of Physics, University of Oviedo, Oviedo, Spain. [2] Center of Research on Nanomaterials and Nanotechnology, CINN (CSIC-Universidad de Oviedo), El Entrego, Spain. [3] Center for Photonics and 2D Materials, Moscow Institute of Physics and Technology, Dolgoprudny, Russia. [4] CIC nanoGUNE BRTA, Donostia–San Sebastian, Spain. [5] Donostia International Physics Center (DIPC), Donostia/San Sebastián, Spain. [6] National Centre for Nano Fabrication and Characterization, Technical University of Denmark, Lyngby, Denmark. [7] DTU, Fotonik, Department of Photonics Engineering and Center for Nanostructured Graphene, Technical University of Denmark, Lyngby, Denmark. [8] Tim Taylor Department of Chemical Engineering, Kansas State University, Manhattan, KS, USA. [9] GrapheneTek, Skolkovo Innovation Center, Moscow, Russia. [10] IKERBASQUE, Basque Foundation for Science, Bilbao, Spain. [11] CIC nanoGUNE BRTA and Department of Electricity and Electronics, UPV/EHU, Donostia - San Sebastian, Spain. [12] These authors contributed equally: J. Duan, G. Álvarez-Pérez. ✉email: pabloalonso@uniovi.es

Hyperbolic electromagnetic waves arise as a consequence of the intrinsic anisotropy of the crystal lattice in natural media[1–3] and of the artificially engineered anisotropic dielectric environment in metamaterials[4–12], which leads to a metallic-like response (negative permittivity) along one (two) of the principal axes in such media and a dielectric-like response (positive permittivity) along the other two (one). Despite their fundamental interest and their potential for the development of new optical applications, these exotic waves are still scarcely explored. In particular, refraction of hyperbolic waves has only been studied for the case in which the incident beam comes from an isotropic medium, typically free space[5–7]. As such, the general case of refraction, involving hyperbolic waves in which both the incident and the refracted waves exhibit non-collinear wavevector **k** and energy flux **S**, remains experimentally unexplored, particularly at the nanoscale, where the specific case of negative refraction of highly confined polaritons has only recently been theoretically proposed[13,14]. The study of the general case of refraction could extend our capabilities to control the flow of light.

Importantly, the recent discoveries of phonon polaritons (PhPs) in van der Waals crystals[15,16] with hyperbolic dispersion, such as h-BN[17–19], α-MoO₃[20–25], and α-V₂O₅[26], have provided unique material platforms to study optical phenomena[27,28] within strongly anisotropic natural media. In particular, PhPs in α-MoO₃ feature in-plane hyperbolic propagation, ultra-low losses, and strong confinement, offering the possibility to visualize refraction directly on the crystal surface and at the nanoscale, which can open new routes in planar nano-optics[29].

Here, we theoretically and experimentally demonstrate the general case of refraction at the interface between two strongly anisotropic (hyperbolic) media. Importantly, we do it at the nanoscale and in a low-loss natural medium by visualizing the propagation of polaritons as they traverse planar nanoprisms tailored on the surface of α-MoO₃. Our images show non-intuitive directions of propagation and strong confinement of the refracted waves, enabling to unveil an exotic optical effect: bending-free refraction, which extends the current capabilities to control the propagation of light at the nanoscale. Furthermore, we develop an in-plane refractive hyperlens, yielding foci as small as $\lambda_p/6$, being $\lambda_p$ the polariton wavelength ($\lambda_0/50$ with respect to the wavelength of light in free space). Our findings provide fundamental knowledge and an effective strategy for the manipulation of polaritons in anisotropic media, paving the way for integrated flat subwavelength optics.

## Results and Discussion

**Theory of refraction between hyperbolic media.** The unique properties of polaritons in hyperbolic media can be better understood by analyzing their isofrequency curve (IFC), a slice of the polariton dispersion surface in momentum-frequency space ($k_x$, $k_y$, $\omega$) by a plane of constant frequency $\omega_0$. The IFCs of polaritons in two different hyperbolic media are illustrated in Fig. 1a, b. For convenience, we consider these two media to be defined by the same hyperbolic slab (with representative permittivity $\varepsilon_x = -5$, $\varepsilon_y = 1$, $\varepsilon_z = 5$, see 'Methods') placed on two different dielectric substrates (with permittivities $\varepsilon_{sub} = 1$ and $\varepsilon_{sub} = 5$). In both cases, the IFCs describe open hyperbolas (black and gray curves, respectively). As a result, not all wavevectors **k** are allowed in these media, which implies that polaritons cannot propagate along all in-plane directions in real space. In fact, propagation is only allowed within the sectors $|\tan(k_x/k_y)| < \sqrt{-\varepsilon_y/\varepsilon_x}$ limited by the asymptotes of the hyperbola in the ($k_x$, $k_y$) plane (see Fig. 1a, b). Additionally, the

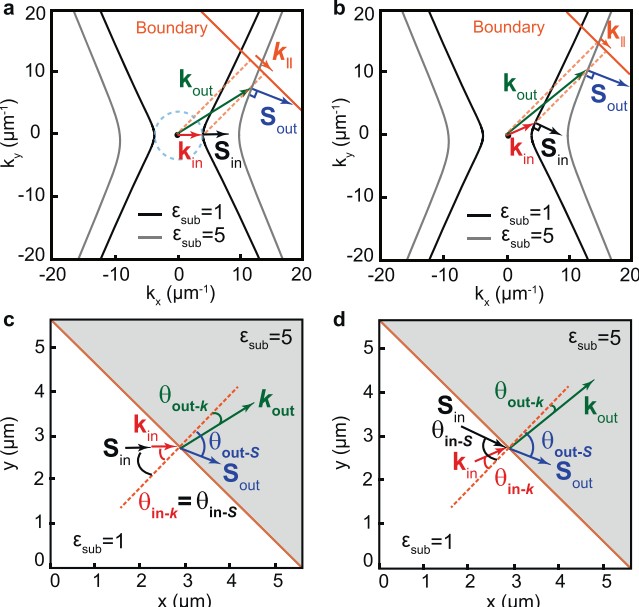

**Fig. 1 Schematics of refraction of polaritons between two hyperbolic media. a** Isofrequency curves of polaritons propagating in a hyperbolic slab (with $\varepsilon_x = -5$; $\varepsilon_y = 1$; $\varepsilon_z = 5$) placed on two different semi-infinite substrates with $\varepsilon_{sub} = 1$ (black curve) and $\varepsilon_{sub} = 5$ (gray curve) that define two different hyperbolic media (medium 1 and 2, respectively). The incident wave in medium 1 is characterized by collinear $k_{in}$ and $S_{in}$ (as in an isotropic medium, indicated by a dashed cyan circle). Upon refraction into medium 2, momentum conservation at the boundary (orange line), $k_{||}$ ($k_{||} = k_{in} \bullet \sin\varphi$, where $\varphi$ is the angle of the boundary), is fulfilled by non-collinear $k_{out}$ and $S_{out}$. The dashed orange lines represent the normal to the boundary. **b** The general case of refraction between two hyperbolic media is represented by an incident wave from medium 1 with non-collinear $k_{in}$ and $S_{in}$ (normal to the isofrequency curve). When the wave refracts into medium 2, momentum conservation at the boundary (orange line) is fulfilled by non-collinear $k_{out}$ and $S_{out}$. The dashed orange lines represent the normal to the boundary. **c** Real-space illustration of refraction between two hyperbolic media shown in **a** where the incident wave exhibits collinear $k_{in}$ and $S_{in}$, i.e. $\theta_{in-k} = \theta_{in-S}$, giving rise to non-collinear $k_{out}$ and $S_{out}$, i.e. $\theta_{out-S} \neq \theta_{out-k}$. **d** Real-space illustration of the general case of refraction between two hyperbolic media shown in **b** where both the incident and the outgoing wave exhibits non-collinear **k** and **S**, i.e. $\theta_{in-k} \neq \theta_{in-S}$ and $\theta_{out-k} \neq \theta_{out-S}$. The tangents parallel to both hyperbolas give rise to bending-free refraction, i.e. $\theta_{in-S} \approx \theta_{out-S}$. The orange dashed lines in **c**, **d** represent the normal to the boundary. The white and gray regions in **c**, **d** correspond to α-MoO₃/air and α-MoO₃/SiO₂, respectively.

Poynting vector **S**—which determines the propagation direction of the polariton and is normal to the IFC[30,31]—is not in general collinear with **k**, as indicated in Fig. 1a (they are collinear only along the x-axis in Fig. 1a). As such, the properties of propagating polaritons in hyperbolic media are different to those in isotropic media, where the IFCs are circular (see dashed cyan curve in Fig. 1a) and polaritons, as is well-known, are allowed to propagate along all in-plane directions in real space with the same absolute value of the wavevector, **k**, which is always collinear to **S**. Importantly, such properties of polaritons propagating in hyperbolic media have a dramatic effect when they refract at a boundary between two different hyperbolic media. Particularly, momentum conservation at the boundary implies that the projection $k_{||}$ of the incident and refracted wavevectors ($k_{in}$ and $k_{out}$, respectively) must be conserved (with $k_{||} = k_{in} \bullet \sin\varphi$, where $\varphi$ is the angle of the boundary as shown in Fig. 1a), giving rise to the

generalized Snell's law[32]:

$$k_{in} \cdot \sin(\theta_{in-k}) = k_{out} \cdot \sin(\theta_{out-k}), \qquad (1)$$

where $\theta_{in-k}$ and $\theta_{out-k}$ are the angles that $\mathbf{k}_{in}$ and $\mathbf{k}_{out}$ form with the normal to the boundary, respectively (see Fig. 1c, d). The propagation directions of the incident and refracted waves are then given by $\mathbf{S}_{in}$ and $\mathbf{S}_{out}$, respectively, i.e. the directions normal to the hyperbolic IFCs for each case, which are in general non-collinear with $\mathbf{k}_{in}$ and $\mathbf{k}_{out}$ (Fig. 1b–d), and thus refraction can occur at angles $\theta_{out-S}$ that can be different from $\theta_{out-k}$. This behavior is in stark contrast to that in isotropic media, in which momentum conservation at the boundary implies that refraction occurs always at $\theta_{out-k}=\theta_{out-S}$.

A particular case of refraction between hyperbolic media for an incident wave with collinear $\mathbf{k}_{in}$ and $\mathbf{S}_{in}$ is shown in Fig. 1a. We observe that the refracted wave ($\mathbf{S}_{out}$) bends away from the direction of $\mathbf{S}_{in}$ towards the boundary (Fig. 1c), in contrast to what is expected in isotropic media for a wave passing from a low refractive index to a high refractive index, where $\mathbf{S}_{out}$ bends towards the normal to the boundary (see Supplementary Fig. 8). Also, the modulus of $\mathbf{k}_{out}$ is much larger than that of $\mathbf{k}_{in}$, showing a strong wavelength reduction.

The general case of refraction occurs when $\mathbf{k}_{in}$ and $\mathbf{S}_{in}$ of the incident wave are not collinear. This case, which has not been tackled experimentally to date, is sketched in Fig. 1d, where a wave impinges at a tilting angle $\theta_{in-S}$ on the boundary between two hyperbolic media. The boundary is also tilted a given angle with respect to the crystal axes. Interestingly, due to the similar shapes of the IFCs in the considered hyperbolic media (α-MoO$_3$/air and α-MoO$_3$/SiO$_2$, black and gray curves in Fig. 1b, respectively), the Poynting vectors of the incident and refracted waves are parallel for almost any $\mathbf{k}_{in}$ and boundary angle $\varphi$, especially in the region where the arms of both hyperbolic IFCs are straight. Hence, the refracted wave propagates almost parallel to the incident wave (i.e. $\theta_{in-S} \approx \theta_{out-S}$), as if the incident wave had been transmitted directly without any change in its propagation direction (black and blue arrows in Fig. 1d). This feature opens the door to the realization of bending-free refraction at arbitrary incident angles in anisotropic media, which is not possible in isotropic media. Note, however, that the modulus and direction of $\mathbf{k}_{out}$ is very different from the modulus and direction of $\mathbf{k}_{in}$, which opens the door to direct engineering of the wavelength and wavefront without modifying the direction of propagation of the wave.

**Nanoimaging of refraction of in-plane hyperbolic polaritons.** We experimentally demonstrate and comprehensively study the characteristics of refraction of polaritons propagating in hyperbolic media. This provides the demonstration of these effects at the nanoscale and/or in a natural medium. To do so, we design and fabricate planar prisms ("Methods") in a slab of the naturally hyperbolic van der Waals crystal α-MoO$_3$. We then visualize the propagation of hyperbolic phonon polaritons (HPhPs) passing through them by polariton wavefront mapping using a scattering-type scanning near-field optical microscope (s-SNOM, see "Methods"). To define the prisms, we etch away triangular regions in a silica (SiO$_2$) substrate on top of which we place a 160-nm-thick α-MoO$_3$ slab, thus forming a region α-MoO$_3$/air with a different refractive index, and thus different polaritonic dispersion, with respect to the region α-MoO$_3$/SiO$_2$. The different polaritonic dispersions in these two regions are clearly corroborated by the near-field image of Fig. 2a, taken at an incident wavelength $\lambda_0 = 11.3\,\mu m$. Specifically, we observe HPhPs launched inside the prism (highlighted by white dashed lines) by the flake edge (see "Methods" and Supplementary Fig. 4), which propagate with collinear $\mathbf{k}_{in}$ and $\mathbf{S}_{in}$ and longer wavelength $\lambda_{in}$ (white arrow), i.e. smaller

wavevector $|\mathbf{k}_{in}| = 2\pi/\lambda_{in}$ (black arrow), than outside the prism ($\lambda_p$, red arrow), thus revealing the lower refractive index of the prisms. We also observe in the same near-field image that when the HPhPs reach a boundary of the prism tilted at an angle $\theta_{in} \sim 55°$, they refract into the α-MoO$_3$/SiO$_2$ region (note that reflection is expected to be very weak, see Supplementary Note 8) along a different direction ($\mathbf{S}_{out-exp}$, blue arrows) with respect to which their wavefronts are tilted ($\mathbf{k}_{out-exp}$, green arrows). Thus, the refracted energy flux and wavevector are not collinear, in agreement with our predictions for hyperbolic refraction in Fig. 1a. Importantly, while the wavevector, $\mathbf{k}_{out-exp}$, refracts towards the normal (note that polariton launching by the prism boundary can be ruled out, see Supplementary Note 3), the energy flux, $\mathbf{S}_{out-exp}$, bends away from it. This is in stark contrast to what is expected for a wave propagating from a lower refractive index region to a higher refractive index region in isotropic media. In addition, the modulus of the refracted wavevector $\mathbf{k}_{out-exp}$ (~6.48 μm$^{-1}$) is considerably larger than that of the incident wavevector $\mathbf{k}_{in}$ (~2.09 μm$^{-1}$) and that of the polariton wavevector along the $x$-direction outside the prism $\mathbf{k}_p$ (~4.08 μm$^{-1}$), revealing a strong wavelength reduction resulting from refraction in hyperbolic media. These findings are in perfect agreement with the case predicted in Figs. 1a–c for hyperbolic refraction considering collinear $\mathbf{k}_{in}$ and $\mathbf{S}_{in}$.

To unambiguously verify the features of refraction between hyperbolic media, we carry out full-wave numerical simulations which mimic our experiments[33] (see "Methods"). The resulting spatial distribution of the out-of-plane component of the electric field, $\text{Re}(E_z(x, y))$, is plotted in Fig. 2b, clearly showing refraction of both the energy flux ($\mathbf{S}_{out}$) and the wavevector ($\mathbf{k}_{out}$), in excellent qualitative and quantitative agreement ($|\mathbf{k}_{out}|$ ~6.35μm$^{-1}$) with the experimental image in Fig. 2a. In addition to numerical simulations, we also validate our experimental results by performing analytical calculations[34] (Fig. 2c) analogous to those shown in Fig. 1. Namely, we calculate the IFCs of HPhPs in α-MoO$_3$/air (gray curve), and α-MoO$_3$/SiO$_2$ (black curve) regions at $\lambda_0 = 11.3\,\mu m$, and, applying the condition of momentum (wavevector) conservation at the boundary (orange line), we extract the Poynting vector and wavevector of the refracted polaritons. Again, we observe refraction of the energy flux ($\mathbf{S}_{out}$, blue arrow) with a tilted wavevector ($\mathbf{k}_{out}$, green arrow) in excellent agreement with the experiment, as well as with the full-wave numerical simulations.

To further analyze refraction in hyperbolic media, we also perform experiments at a different illuminating wavelength $\lambda_0$ (different tilting angles of the prism boundary $\theta$ are also shown in Supplementary Fig. 2), as shown in Fig. 2d–f for $\lambda_0 = 11.1\,\mu m$. Interestingly, we observe that, in this case, both the angular separation between $\mathbf{k}_{out-exp}$ and $\mathbf{S}_{out-exp}$ and the confinement effect are larger, being the refracted wave ($\mathbf{S}_{out-exp}$) almost parallel to the prism boundary and the modulus of $\mathbf{k}_{out-exp}$ (12.47 μm$^{-1}$) about four times larger than that of $\mathbf{k}_{in}$ (~3.14 μm$^{-1}$) and 2.1 times larger than that of $\mathbf{k}_p$ (~5.92 μm$^{-1}$).

Altogether, these results demonstrate the efficient refractive nature of our planar prisms, enabling us to visualize in real-space three important features of highly confined polaritons refracted at the boundary between two hyperbolic media: (i) large tilting of their wavefronts (given by $\mathbf{k}_{out-exp}$) with respect to their propagation direction (given by $\mathbf{S}_{out-exp}$), (ii) counter-intuitive directions of propagation, and (iii) subwavelength confinement (with respect to polaritons along the same crystal axis in the same medium).

**Sub-diffractional planar lensing of hyperbolic polaritons.** Such unique features of refracted polaritons in naturally in-plane hyperbolic media open the door to focus ultra-confined

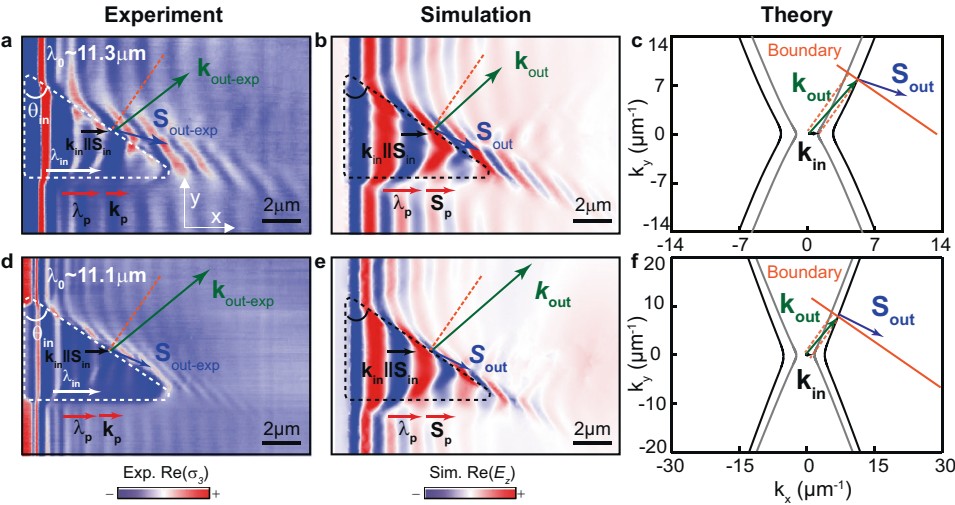

**Fig. 2 Real-space visualization of refraction between two anisotropic media using highly confined polaritons with collinear incident k and S. a, b** Experimental $Re(\sigma_3(x,y))$ (**a**) and simulated $Re(E_z(x,y))$ (**b**) near-field images of HPhPs propagating in a 160-nm-thick α-MoO$_3$ flake at $\lambda_0 = 11.3\,\mu m$. The white and black dashed lines mark triangular prisms fabricated by etching an air cavity on the SiO$_2$ substrate below the α-MoO$_3$ flake. $S_{in}$ and $k_{in}$ display the direction of propagation and the wavevector of incident polaritons in α-MoO$_3$/air, respectively. Horizontal propagation of non-refracted polaritons is marked as $k_p$ and $S_p$. Upon refraction at a boundary of the prism with an angle $\theta_{in} \sim 55°$, HPhPs bend away from the normal, $S_{out-exp}$ (blue arrow), with a tilted wavevector $k_{out-exp}$ (green arrow). Compared to non-refracted HPhPs, indicated by $\lambda_p$, the refracted HPhPs are stronger confined (with a wavelength about 1.6 times shorter). **c** Analytic IFCs of α-MoO$_3$/SiO$_2$ (black hyperbolas) and α-MoO$_3$/air (gray hyperbolas) effective media in **a, b**, and considering momentum conservation at the boundary (orange line), the extracted wavevector and direction of the refracted polaritons, $k_{out}$ and $S_{out}$, respectively, are in good agreement with both experiment and simulation. **d, e** Experimental $Re(\sigma_3(x,y))$ (**d**) and simulated $Re(E_z(x,y))$ (**e**) near-field images of HPhPs propagating in a 160-nm-thick α-MoO$_3$ flake at $\lambda_0 = 11.1\,\mu m$. The refracted HPhPs propagate almost parallel to the boundary with a wavelength 2.1 times smaller than $\lambda_p$. **f** Analytic IFCs of α-MoO$_3$/SiO$_2$ (black hyperbolas) and α-MoO$_3$/air (gray hyperbolas) effective media in **d, e**. The orange dashed lines in **a–f** represent the normal to the boundary.

polaritons in a planar geometry. To demonstrate this possibility, we design and fabricate a planar lens in α-MoO$_3$ (Fig. 3a). As noted above (Fig. 2), when HPhPs with collinear $k_{in}$ and $S_{in}$ (red arrow in Fig. 3b) refract at a boundary with another hyperbolic medium with higher refractive index (such as when passing from α-MoO$_3$/air to α-MoO$_3$/SiO$_2$), they bend away from the normal to the boundary (blue arrow in Fig. 3b). This means that, in the case of considering a prism of triangular shape, as that shown in Fig. 3a, all the refracted polaritons (blue arrows) can converge into a single spot, and thus the prism acts as a focusing lens for highly confined polaritons (see Supplementary Note 9 for the analytical description of general shapes of lenses focusing hyperbolic waves). More importantly, as such a lens is based on refraction of HPhPs, the refracted waves can potentially feature infinitely large wavevectors when the boundary is perpendicular to the asymptote of the hyperbolic IFC, which would yield deeply sub-diffractional foci sizes. However, as HPhPs decay exponentially, the intensity at a distant focus can be weak due to propagation losses, which would be more notable for large wavevectors approaching the asymptote of the hyperbolic IFC. Consequently, we designed our triangular lens looking for a compromise between a large refracted wavevector and a long propagation length of the refracted polaritons. According to our theoretical calculations (Supplementary Fig. 7), this compromise is obtained for an angle of the in-plane wavevector of about 62°. Therefore, we fabricate a triangular prism with boundaries perpendicular to this angle (Fig. 3a). The experimental and simulated near-field images for this lens design upon illumination at 11.16 μm are shown in Figs. 3c, d, respectively. In both images, we observe refraction of incident HPhPs with collinear $k_{in}$ and $S_{in}$ (red arrows) at the lens boundaries (black dashed contour) resulting in HPhPs with non-collinear $k_{out}$ and $S_{out}$ (green and blue arrows, respectively) propagating along directions almost parallel to the

boundaries (blue arrows), which eventually converge, resulting in a focus. This result is in stark contrast to that observed in a similar lens based on refraction of highly confined polaritons in an isotropic material, such as h-BN, in which refracted polaritons bend towards the normal (Supplementary Fig. 8), making all them to diverge (Fig. 3e). The near-field images of our hyperbolic lens reveal that the wavevector $k_{out}$ of the refracted HPhPs is much larger than the wavevector $k_{in}$ of the incident HPhPs (note that in the experimental image there is a contribution of tip-launched HPhPs with two different $k_{in}$ wavevectors). More importantly, the wavevector $k_{out}$ is also much larger than the wavevector $k_p$ of HPhPs propagating along the x-direction in the flake on top of SiO$_2$ (black arrows). In order to avoid the influence from the permittivity of substrate, we evaluate the focusing resolution by comparing the full-width at half-maximum (FWHM) with the wavelength of polaritons propagating in MoO$_3$/SiO$_2$ ($\lambda_p = 2\pi/|k_p|$). Since the focus shows a FWHM of ~240 nm (red dots and gray curve in Fig. 3f, for experimental and simulated line profiles, respectively), we obtain a focus that is much smaller than the polariton wavelength ($\lambda_p$) along the x-direction, or the free-space illumination ($\lambda_0$), namely of ~$\lambda_p$/6, or $\lambda_0$/50. This resolution reveals that a diffraction-limited optical system in hyperbolic media can show a focus that is much smaller than the incident polaritonic wavelength, which again reflects the unique behavior of electromagnetic waves in hyperbolic media. Moreover, this result confirms our planar lens based on refraction of HPhPs as a nano-optical element that greatly exceeds the focusing resolution of any lens based on refraction of highly confined polaritons in isotropic media[35–38].

**Visualization of the general case of refraction.** So far, we have visualized refraction in hyperbolic media for the case in which

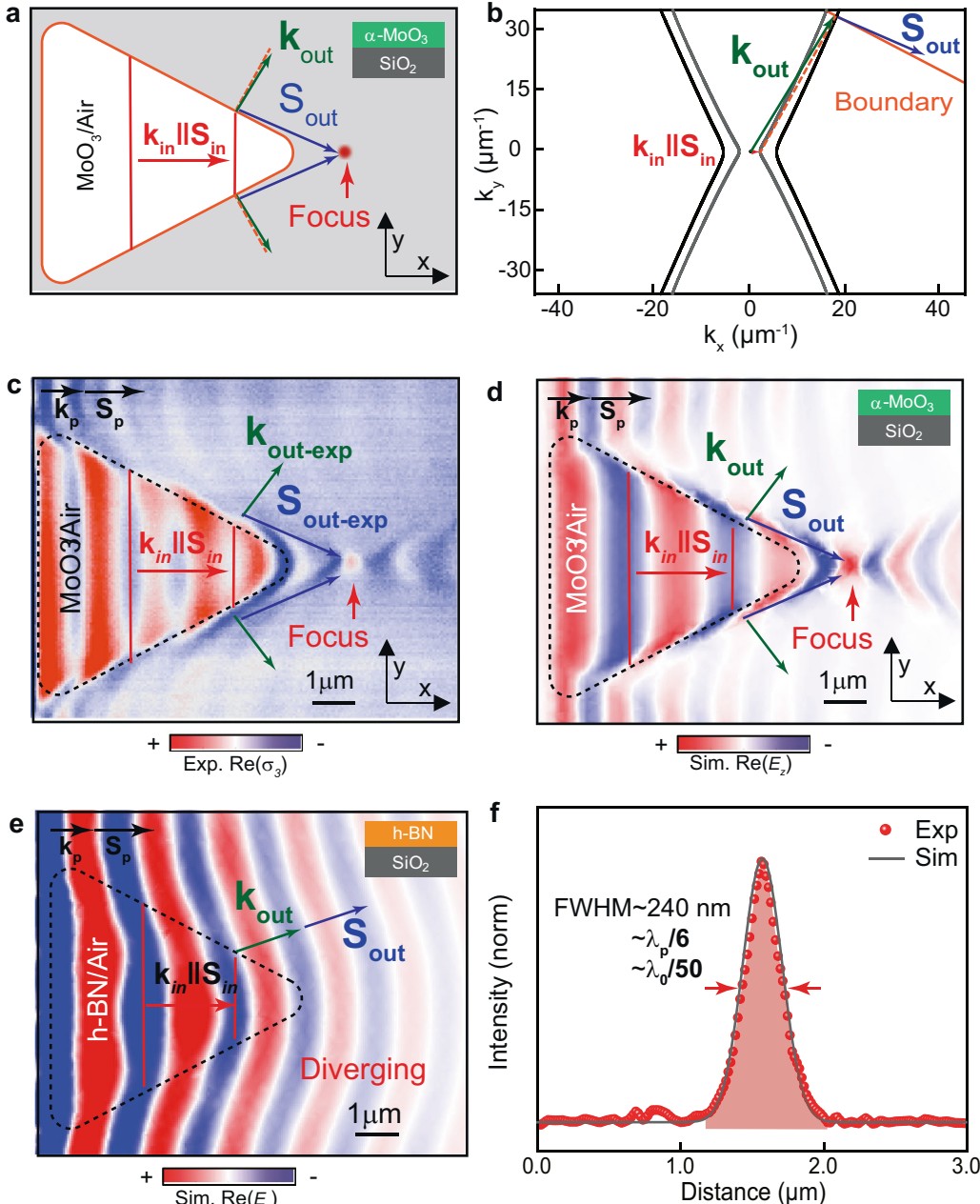

**Fig. 3 Sub-diffractional planar lens based on refraction of HPhPs. a** Schematics of a refractive hyperbolic lens fabricated by etching a triangular air cavity on the SiO$_2$ substrate below the α-MoO$_3$ flake. The top and bottom boundaries of the lens have the same slope as the boundary in **b**. Upon refraction at the boundaries (orange contour line), polaritons bend far away from the normal, **S**$_{out}$ (blue arrows), with a tilted wavevector **k**$_{out}$ (green arrows), converging at a focal spot (red dot). **S**$_{in}$ and **k**$_{in}$ display the direction of propagation and the wavevector of incident polaritons in α-MoO$_3$/air, respectively. **b** Analytic isofrequency curves (IFCs) of polaritons propagating in α-MoO$_3$/SiO$_2$ (black hyperbola) and α-MoO$_3$/air (gray hyperbola). When the boundary (orange line) is nearly perpendicular to the asymptote of the open hyperbolic IFC, the refracted polaritons propagate (**S**$_{out}$, blue arrows) almost parallel to the boundary with large non-collinear wavevector (**k**$_{out}$, green arrows). **c** Experimental near-field image of the refractive planar hyperlens (black dashed line) for polaritons in a 170-nm-thick α-MoO$_3$ slab, at $\lambda_0 = 11.16$ μm. The polaritons converge upon refraction at the triangular boundary. Compared to non-refracted polaritons, indicated by **k**$_p$ and **S**$_p$ (black arrows) the refracted polaritons, **S**$_{out-exp}$ (blue arrows), propagate nearly parallel to the boundary. **d** Simulated near-field image of the refractive planar hyperlens (black dashed line) considered in **b** and visualized in **c**. **e**, Simulated near-field image of a refractive lens for in-plane isotropic polaritons in a 170-nm-thick h-BN slab, at $\lambda_0 = 6.5$ μm. Upon refraction at the triangular boundary (black dashed line), polaritons bend towards the normal, **S**$_{out}$ (blue arrow), with collinear wavevector **k**$_{out}$ (green arrow), yielding a diverging effect. Horizontal propagation of non-refracted polaritons is marked as **k**$_p$ and **S**$_p$. **f** Near-field intensity profiles (gray line and red dots) extracted through the focus spot along the vertical direction in **d** and **c**, respectively. Both curves are normalized to the near-field intensity far away from the lens and flake edges. Confinement factors as large as ~$\lambda_p/6$ and ~$\lambda_0/50$ are obtained with respect to the polariton and free-space light wavelengths.

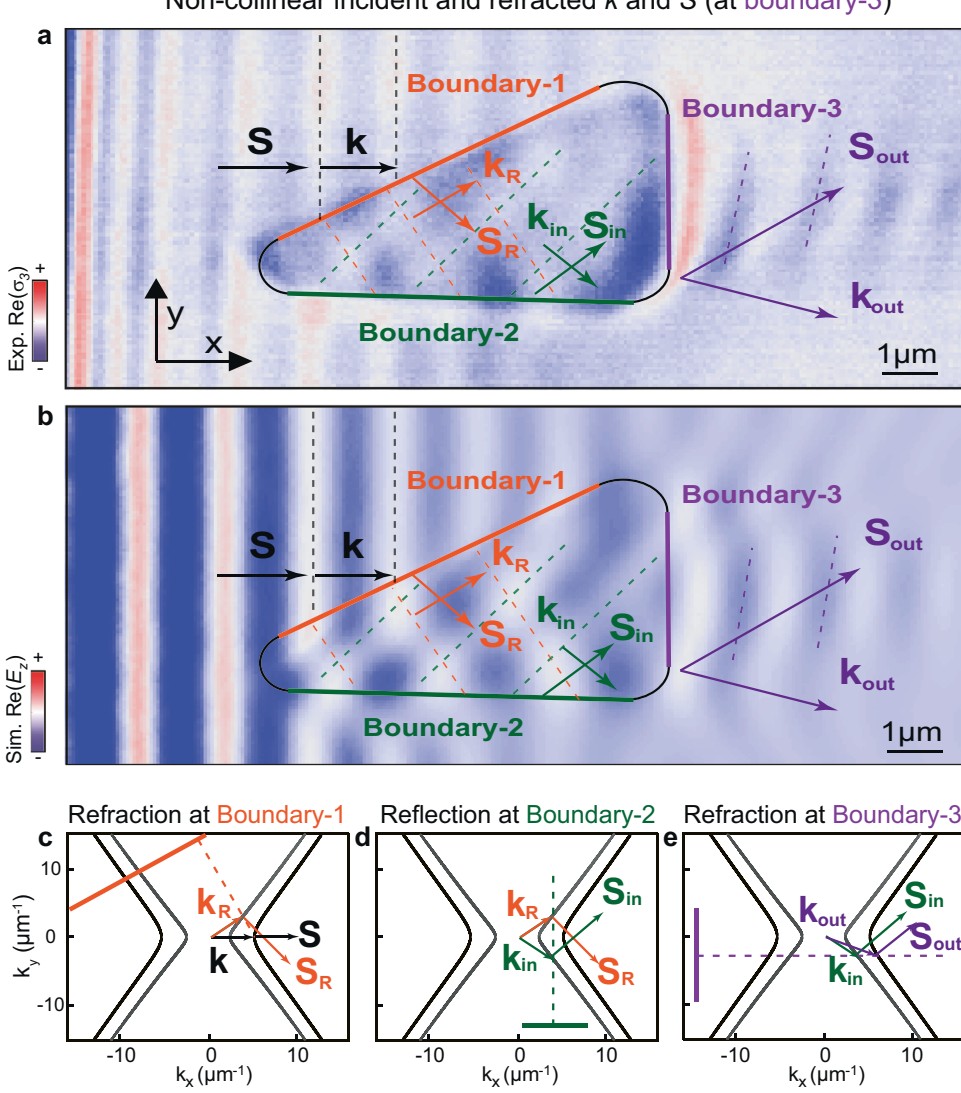

**Fig. 4 Real-space visualization of the general case of refraction between two anisotropic media using nanoscale-confined HPhPs passing through a bending-free planar prism. a** Experimental $\mathrm{Re}\left(\sigma_3(\mathbf{x}, \mathbf{y})\right)$ near-field images of polaritons propagating in a 231-nm-thick α-MoO₃ flake at $\lambda_0 = 11.0\ \mu m$. The black contour line marks a triangular prism fabricated by etching an air cavity on the SiO₂ substrate below the α-MoO₃ flake. A first refraction takes place at boundary-1 (orange solid line) for incident polaritons with collinear **k** and **S** (black arrows), yielding refracted polaritons with non-collinear $\mathbf{k}_R$ and $\mathbf{S}_R$ (orange arrows). These polaritons then reflect at boundary-2 (green solid line), yielding polaritons with non-collinear $\mathbf{k}_{in}$ and $\mathbf{S}_{in}$ (green arrows). A second refraction at boundary-3 (violet solid line) yields polaritons with non-collinear $\mathbf{k}_{out}$ and $\mathbf{S}_{out}$ (violet arrows). **b** Simulated $\mathrm{Re}(E_z(x, y))$ near-field images of HPhPs for the case shown in **a**. Dashed lines in experimental and simulated near-field images indicate the wavefronts of polaritons as they pass through the prisms. **c–e** Analytic IFCs of HPhPs in MoO₃/air (gray curve) and MoO₃/SiO₂ (black curve), predicting the directions of refraction or reflection of HPhPs at boundary-1 (**c**), boundary-2 (**d**), and boundary-3 (**e**) based on momentum conservation. The orange (**c**), green (**d**), and purple (**e**) solid lines represent the boundary-1, boundary-2, and boundary-3, while the corresponding dashed lines represent the normal to the boundary.

collinear $\mathbf{k}_{in}$ and $\mathbf{S}_{in}$ refract into polaritons with non-collinear $\mathbf{k}_{out}$ and $\mathbf{S}_{out}$. However, the general phenomenon of refraction involves the incident polaritons exhibiting non-collinear $\mathbf{k}_{in}$ and $\mathbf{S}_{in}$ (as sketched in Fig. 1b, d). In the following, we study this fundamental phenomenon in hyperbolic media (Fig. 4). To do this, we again fabricate prisms in α-MoO₃ (following the same structure design as in Fig. 2) and visualize (by s-SNOM) the propagation of HPhPs refracting upon them. As shown in the near-field image of Fig. 4a, we observe that HPhPs launched by the edge of the flake (black arrows) refract at boundary-1 (Fig. 4c), and the outcoming HPhPs propagate with non-collinear $\mathbf{k}_R$ and $\mathbf{S}_R$ inside the prism (orange arrows). As such, these polaritons can now be used to visualize the general case of refraction at another boundary of the prism. However, to carry out this

experiment successfully, we need to ensure that: (i) the angle of the boundary allows refraction of polaritons according to momentum conservation, and (ii) the HPhPs can reach this boundary within a reasonable propagation distance. We fulfill these conditions by considering a horizontal boundary (boundary-2) followed by a vertical boundary (boundary-3) in the triangular prism, as shown in the near-field image of Fig. 4a. In particular, we observe that HPhPs with non-collinear $\mathbf{k}_R$ and $\mathbf{S}_R$ are reflected at boundary-2 (Fig. 4d), yielding polaritons with non-collinear $\mathbf{k}_{in}$ and $\mathbf{S}_{in}$ (green arrows) propagating directly towards boundary-3, and reaching it within a reasonably short distance. Consequently, these polaritons refract at boundary-3, which results in polaritons with non-collinear $\mathbf{k}_{out}$ and $\mathbf{S}_{out}$ (violet arrows), as predicted by momentum conservation (Fig. 4e), and

in good agreement with numerical simulations mimicking the experiment (Fig. 4b). This result unambiguously demonstrates refraction of waves whose energy flux and wavevector directions are non-collinear, and thus constitutes the real-space visualization of the general case of refraction between two hyperbolic media, which, in addition, we demonstrate at the nanoscale and in a natural medium. Furthermore, we highlight that, since $\mathbf{k}_{in}$ and $\mathbf{k}_{out}$ are close to the asymptotes of the IFCs, where the tangents to both hyperbolas are parallel, $\mathbf{S}_{in}$ and $\mathbf{S}_{out}$ are almost parallel after refraction at boundary-3. As a result, polaritons refract upon boundary-3 behave as if they had been directly transmitted without any change in their direction of propagation, despite the wavevector does change upon this refraction phenomenon. Therefore, refraction upon this prism is bending-free, in excellent agreement with our theoretical prediction shown in Fig. 1d. Such bending-free refractive prism showed in our work will open possibilities to engineer polaritonic wavefronts at the nanoscale without the need of changing their direction of propagation.

## Conclusions

In summary, our work explores the character of refraction between two strongly anisotropic media, which, despite its fundamental importance, has remained elusive up to now. Our observations of refraction of strongly confined polaritons in a hyperbolic biaxial van der Waals crystal reveal an exotic optical effect: bending-free refraction, which opens the door to on-demand steering of light at the nanoscale and in natural media. Furthermore, our demonstration of a subwavelength planar lens at the nanoscale based on hyperbolic refraction paves the way for the development of planar nano-optical elements in anisotropic media. Altogether, our results open new avenues for integrated flat optics, directional energy transfer, and heat management applications, as well as for mid-infrared (bio) sensing.

## Methods

**Sample fabrication and characterization**. The α-MoO$_3$ and h-BN nanometer-thick flakes were obtained by mechanical exfoliation using a Nitto tape (Nitto Denko Co., SPV 224P) from commercial α-MoO$_3$ bulk crystals and isotopically enriched h-BN ($^{10}$B) crystals grown via the atmospheric pressure flux growth method[19]. First, the bulk crystals were thinned down employing the Nitto tape and transferred on an optically transparent polydimethylsiloxane (PDMS) stamp. Second, selected flakes with sharp edges were identified by an optical microscope and transferred on the target substrate by the dry-transfer technique, which allows a precise positioning and alignment of the flakes on top of the triangular air regions (2D prisms) fabricated on a 500-nm-thick SiO$_2$ layer grown by wet oxidation on a Si substrate. For an efficient transfer, the substrate was heated up to 200 °C[39].

The triangular air regions were fabricated in a two-steps process: (i) the air structured patterns were defined using a direct writing system (MLA 100, Heidelberg Instrument) equipped with a 365 nm LED light source on a 1.5-μm-thick positive tone photoresist (AZ MiR 701, MicroChemicals) previously deposited on the SiO$_2$/Si substrate; (ii) part of the SiO$_2$ layer was etched away by a fluorine-based plasma using an inductively coupled plasma system (Advanced Oxide Etcher, SPTS), generating gaps with a depth of around 350 nm into the SiO$_2$ layer. To get rid of photoresist residues and contaminants, the etched substrates were cleaned with a plasma ashing system (PVA TePla 300, PVA TePla AG) before transferring α-MoO$_3$ and h-BN flakes.

**Infrared near-field nanoimaging**. Infrared nanoimaging was performed with a commercial scattering-type scanning near-field optical microscope (s-SNOM)[40,41] from Neaspec GmbH. A tunable DRS Daylight Solutions quantum cascade laser (from 880 to 1100 cm$^{-1}$) was used as excitation source by focusing the light onto a metal-coated (Pt/Ir) AFM (atomic force microscope) tip oscillating at a tapping frequency of ~280 kHz with a tapping amplitude ~100 nm. We illuminate the α-MoO$_3$ flakes with s-polarized infrared light and keep the polarization direction of the incident electric field ($E_{inc}$) perpendicular to the flake edges to launch HPhPs. The flakes were raster scanned and the tip-scattered field $E_{sca}$ was recorded with a pseudo-heterodyne Michelson interferometer[35] and detected using a liquid nitrogen cooled HgCdTe (MCT) detector. To suppress far-field background signals, the detected signal was demodulated at the $n$th harmonics of the tip oscillating frequency ($n = 3$ in our work), yielding the complex-valued near-field signals $\sigma_n = s_n e^{i\varphi_n}$, with $s_n$ being the near-field amplitude and $\varphi_n$ being the near-field

phase. Throughout the manuscript, we show the real part of the near-field signal, Re($\sigma_3(x, y)$), as a function of the tip position $(x, y)$. Edge-launched polaritons yield fringes with spacing $\lambda_p$, where $\lambda_p$ is the polaritonic wavelength, while tip-launched polaritons produce fringes with period of $\lambda_p/2$, existing close to the flake edges (see Figs. 2 and 3 in the main text).

**Full-wave numerical simulations using finite-element methods**. We simulated s-SNOM near-field images using the finite-element-method numerical software COMSOL MULTIPHYSICS. α-MoO$_3$ slabs were placed on top of SiO$_2$ substrates where we defined the same geometry of the air structures as in the experiments. The thickness of the slabs was set to the value extracted from AFM measurements of the α-MoO$_3$ flakes in the corresponding experimental s-SNOM images. The whole system was illuminated with plane waves with s-polarization perpendicular to the flake edges. Given that the s-SNOM signal can be approximated by the vertical component of the electric field[42], we calculated the real part of $z$ component the of the near-field signal at 50 nm above the slab surface, Re$(E_z(x, y))$. Meshing types and sizes were optimized to ensure a good convergence of simulated results. The experimental s-SNOM images were well reproduced by our simulated images (see Figs. 2–4 in the main text). Yet, since our numerical simulations do not consider the influence of the tip, only the wavefront of edge-launched polaritons is seen (note the difference with the measurements close to the flake edges, where dense fringes appear due to tip-launched polaritons). The dielectric permittivity for α-MoO$_3$ and isotopically enriched h-BN were taken from references[19,33]. The real part of the dielectric function of SiO$_2$ used throughout this work was extracted by fitting the values reported in references[43–45] to experimental HPhPs dispersions with a Lorentzian model.

**HPhPs dispersion and IFC from analytical calculations**. The analytically calculated dispersion for polaritons in a biaxial slab embedded between two semi-infinite media[34] is given by

$$k(\omega) = \frac{\rho}{d} \left[ \arctan\left( \frac{\varepsilon_1 \rho}{\varepsilon_z} \right) + \arctan\left( \frac{\varepsilon_3 \rho}{\varepsilon_z} \right) + \pi l \right], l = 0, 1, 2 \dots ,$$

where $k$ is the in-plane momentum, i.e. $k = \sqrt{k_x^2 + k_y^2}$, $\varepsilon_1$ and $\varepsilon_3$ are the permittivity of the superstrate and substrate, respectively, $d$ is the thickness of biaxial slab, $\rho = i\sqrt{\varepsilon_z/(\varepsilon_x \cos^2\alpha + \varepsilon_y \sin^2\alpha)}$, and where $\alpha$ is the angle between the $x$-axis and $\mathbf{k}$. Based on the equation above, we calculate the IFC of HPhPs by varying $\alpha$ from 0° to 360° for a fixed incident frequency $\omega$. In all cases we show propagating modes, i.e. those for which |Re($k$)|>|Im($k$)|. In addition, since the polaritonic wavevector depends on $\varepsilon_3$ ($\varepsilon_1 = 1$ all throughout this work as the superstrate is air), different substrate permittivities can give rise to different effective hyperbolic media.

For the general case discussed in Fig. 1 in the main text, we select representative permittivity values: $\varepsilon_x = \varepsilon_y = \varepsilon_z = -3$ for the isotropic slab and $\varepsilon_x = -5$, $\varepsilon_y = 1$, $\varepsilon_z = 5$ for the hyperbolic slab, respectively, since at least one negative component is necessary for the existence of polaritons. The slab thickness is set to 100 nm in both cases. The permittivity of the substrates $\varepsilon_3$ are set to 1 and 5, for the medium in which polaritons are incident and refracted, respectively (white and gray regions in Fig. 1c, d).

For the specific case discussed in Figs. 2–4 we calculate the ICFs in α-MoO$_3$ slabs on top of air and SiO$_2$ using the permittivity values described in the section "Full-wave numerical simulations using finite-element methods". The thickness of α-MoO$_3$ flakes was set to the value extracted from topography measurements.

## Data availability

All data that support the findings of this study are available from the corresponding author upon reasonable request

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

## Acknowledgements

G.Á.-P. and J.T.-G. acknowledge support through the Severo Ochoa Program from the Government of the Principality of Asturias (nos. PA-20-PF-BP19-053 and PA-18-PF-BP17-126, respectively). S.X. acknowledges the support from Independent Research Fund Denmark (Project No. 9041-00333B). B.C. acknowledges the support from VILLUM FONDEN (No. 00027987). The Center for Nanostructured Graphene is sponsored by the Danish National Research Foundation (Project No. DNRF103.) K.V.V. and V.S.V. gratefully acknowledge the financial support from the Ministry of Science and Higher Education of the Russian Federation (Agreement No. 075-15-2021-606). J.M.-S. acknowledges financial support through the Ramón y Cajal Program from the Government of Spain (RYC2018-026196-I). A.Y.N. and J.I.M. acknowledge the Spanish Ministry of Science, Innovation and Universities (national projects MAT201788358-C3-3-R and PID2019-104604RB/AEI/10.13039/501100011033). R.H. acknowledges financial support from the Spanish Ministry of Science, Innovation and Universities (national project RTI2018-094830-B-100 and the project MDM-2016-0618 of the Marie de Maeztu Units of Excellence Program) and the Basque Government (grant No. IT1164-19). A.Y. N. also acknowledges the Basque Department of Education (grant no. PIBA-2020-1-0014). P.A.-G. acknowledges support from the European Research Council under starting grant no. 715496, 2DNANOPTICA and the Spanish Ministry of Science and Innovation (State Plan for Scientific and Technical Research and Innovation grant number PID2019-111156GB-I00).

## Author contributions

P.A.-G. and J.D. conceived the study. P.A.-G. supervised the project. J.D. carried out the near-field imaging measurements with the help of A.B. J.D., A.I.F.T.-M., S.X., B.C., and J.I.M. contributed to sample fabrication. S.L. and J.H.E. provided the isotopically enriched boron nitride. G.Á.-P. and K.V.V. carried out the analytical calculations and A.I.F. T.-M. performed the numerical simulations with the help of G.Á.-P. and supervised by J.M.-S. and A.Y.N. P.A.-G., J.D., G.Á.-P., J.T.-G., V.S.V., A.Y.N., R.H., and J.M.-S. participated in data analysis. P.A.-G., J.D., and G.Á.-P. wrote the manuscript with input from the rest of authors.

## Competing interests

R.H. is cofounder of Neaspec GmbH, a company producing scattering-type near-field scanning optical microscope systems, such as the one used in this study. The remaining authors declare no competing interests.
