## [Peer Review File · Nature Communications]

REVIEWER COMMENTS

Reviewer #1 (Remarks to the Author):

I'd like to thank the authors for the detailed response. My concerns have been addressed and the paper is quite suitable for Nature Communications now.

Reviewer #2 (Remarks to the Author):

This work is one of the clearest direct demonstrations of the unusual characteristics of refraction in anisotropic media, made possible by the in-plane anisotropy and low loss of hyperbolic polaritons in MoO₃. The extremely tight focusing of the hyperlens is especially impressive. This work will be helpful to other near-field microscopists interpreting their own images and may inspire development of future devices. I can recommend this manuscript for publication in Nature Communications after the following changes are made:

In the second paragraph, "metallic-like response along one of the optical axes". The authors meant to say "principal" axes, not "optical/optic" axes. These are different things. Principal axes form the basis that diagonalizes the dielectric tensor. The optic axes are special directions along which unpolarized light does not experience birefringence.

The phrase "isotropic media, where the IFCs describe circumferences" is poorly worded. Consider rephrasing as "isotropic media, where the IFCs are circular."

Define k_{\parallel} mathematically. This "momentum conservation" line changes to red in Fig. 2 although it is still referred to as "orange." Also, what are the orange, green, and purple bars in Fig. 4c-e?

These are not mentioned in the caption.

Fig. 2 should be split into more than two subfigures.

It is a stretch to refer to anisotropic refraction as "anomalous" considering that it is commonplace (just difficult to image directly). Remove this terminology? In the literature, "anomalous refraction" refers to more exotic phenomena. See for example D. Sell, et al. ACS Photonics 5 (2018) 2402-2407

In Fig. 4 caption title and in other parts of the text: "the most general case of refraction" can just read "general case of refraction". Likewise, in the second to last paragraph, "most extreme anisotropic case" can just be "extreme anisotropic case." Many instances of using "very" in the main text and SOM can be removed. The authors should avoid other unnecessary superlatives.

The term "diffraction-free" lensing appears in the title but is never explained.

There are some problems with the "bending-free refraction" part of this work:

Throughout the text, the authors give the impression that "bending-free" refraction is only possible in hyperbolic media because of their "unique properties", only to finally say that it is also possible in non-hyperbolic anisotropic media in the second-to-last paragraph. The explanation of why this occurs seems to rely on the asymptotes of the hyperbola, so it is unclear why this would be a more general phenomenon. Furthermore, the statement "IFCS... becomes two parallel straight lines" does not make sense. The IFCs are not changing, the k-vectors are approaching the IFC asymptotes. The author should provide a better general intuitive understanding of why bending-free refraction occurs.

Response to Referee #1:

I'd like to thank the authors for the detailed response. My concerns have been addressed and the paper is quite suitable for Nature Communications now.

We thank the referee for the positive assessment of our work and his/her recommendation for its publication in Nature Communications.

Response to Referee #2:

This work is one of the clearest direct demonstrations of the unusual characteristics of refraction in anisotropic media, made possible by the in-plane anisotropy and low loss of hyperbolic polaritons in MoO₃. The extremely tight focusing of the hyperlens is especially impressive. This work will be helpful to other near-field microscopists interpreting their own images and may inspire development of future devices. I can recommend this manuscript for publication in Nature Communications after the following changes are made:

We thank the referee for the positive evaluation of our work and his/her valuable comments. In the revised version we have carefully addressed all the concerns raised, which definitely have helped to improve the quality of our manuscript. We hope the revised manuscript can be satisfactory for the referee to recommend its publication in Nature Communications.

-In the second paragraph, “metallic-like response along one of the optical axes”. The authors meant to say “principal” axes, not “optical/optic” axes. These are different things. Principal axes form the basis that diagonalizes the dielectric tensor. The optic axes are special directions along which unpolarized light does not experience birefringence.

We agree with the referee and thank him/her for the careful checking of the manuscript. We have replaced the term of “optical axes” by “principal axes” in the revised manuscript.

-The phrase “isotropic media, where the IFCs describe circumferences” is poorly worded. Consider rephrasing as “isotropic media, where the IFCs are circular.”

We agree with the referee and have modified the sentence (in red) in the revised manuscript:

“As such, the properties of propagating polaritons in hyperbolic media are very different to those in isotropic media, **where the IFCs are circular** (see dashed cyan curve in Fig. 1a) ...”

-Define k_{\parallel} mathematically. This “momentum conservation” line changes to red in Fig. 2 although it is still referred to as “orange.” Also, what are the orange, green, and purple bars in Fig. 4c-e? These are not mentioned in the caption.

We thank the referee for this valuable comment. In the revised manuscript, we have defined $k_{\parallel} = k_{in} \cdot \sin \varphi$, where φ is the angle of the boundary. In addition, we have changed the “momentum conservation” line to orange in Fig. 2 and have included the description of the orange, green and purple bars in the caption of Fig. 4.

Fig. 2 | Real-space visualization of refraction between two anisotropic media using highly confined polaritons with collinear incident k and S . **a-b**, Experimental $\text{Re}(\sigma_3(x, y))$ (a) and simulated $\text{Re}(E_z(x, y))$ (b) near-field images of HPhPs propagating in a 160-nm-thick α -MoO₃ flake at $\lambda_0 = 11.3 \mu\text{m}$. The white dashed line marks a triangular prism fabricated by etching an air cavity on the SiO₂ substrate below the α -MoO₃ flake. Upon refraction at a boundary of the prism with an angle $\theta_{\text{in}} \sim 55^\circ$, HPhPs bend away from the normal, $S_{\text{out-exp}}$ (blue arrow), with a tilted wavevector $k_{\text{out-exp}}$ (green arrow). Compared to non-refracted HPhPs, indicated by λ_p , the refracted HPhPs are stronger confined (with a wavelength about 1.6 times shorter). **c**, Analytic IFCs of α -MoO₃/SiO₂ (black hyperbolas) and α -MoO₃/air (grey hyperbolas) effective media in (a-b), and considering momentum conservation at the boundary (orange line), the extracted wavevector and direction of the refracted polaritons, k_{out} and S_{out} , respectively, are in good agreement with both experiment and simulation. **d-e**, Experimental $\text{Re}(\sigma_3(x, y))$ (d) and simulated $\text{Re}(E_z(x, y))$ (e) near-field images of HPhPs propagating in a 160-nm-thick α -MoO₃ flake at $\lambda_0 = 11.1 \mu\text{m}$. The refracted HPhPs propagate almost parallel to the boundary with a wavelength 2.1 times smaller than λ_p . **f**, Analytic IFCs of α -MoO₃/SiO₂ (black hyperbolas) and α -MoO₃/air (grey hyperbolas) effective media in (d-e).

Fig. 4 | Real-space visualization of the most general case of refraction between two anisotropic media using nanoscale-confined HPhPs passing through a bending-free planar prism. a, Experimental $\text{Re}(\sigma_3(x, y))$ near-field images of polaritons propagating in a 231-nm-thick α -MoO₃ flake at $\lambda_0 = 11.0 \mu\text{m}$. The black contour line marks a triangular prism fabricated by etching an air cavity on the SiO₂ substrate below the α -MoO₃ flake. A first refraction takes place at boundary-1 (orange solid line) for incident polaritons with collinear \mathbf{k} and \mathbf{S} (black arrows), yielding refracted polaritons with non-collinear \mathbf{k}_R and \mathbf{S}_R (orange arrows). These polaritons then reflect at boundary-2 (green solid line), yielding polaritons with non-collinear \mathbf{k}_{in} and \mathbf{S}_{in} (green arrows). A second refraction at boundary-3 (violet solid line), yields polaritons with non-collinear \mathbf{k}_{out} and \mathbf{S}_{out} (violet arrows). **b**, Simulated $\text{Re}(E_z(x, y))$ near-field images of HPhPs for the case shown in (a). Dashed lines in experimental and simulated near-field images indicate the wavefronts of polaritons as they pass through the prisms. **c-e**, Analytic IFCs of HPhPs in MoO₃/air (grey curve) and MoO₃/SiO₂ (black curve), predicting the directions of refraction or reflection of HPhPs at boundary-1 (c), boundary-2 (d) and boundary-3 (e) based on momentum conservation. **The orange (c), green (d) and purple (e) solid lines represent the boundary-1, boundary-2 and boundary-3, respectively.**

-Fig. 2 should be split into more than two subfigures.

We have rearranged Fig. 2 into six subfigures (Figs. 2a-2f).

-It is a stretch to refer to anisotropic refraction as “anomalous” considering that it is commonplace (just difficult to image directly). Remove this terminology? In the literature, “anomalous refraction” refers to more exotic phenomena. See for example D. Sell, et al. ACS Photonics 5 (2018) 2402-2407

We agree with the referee that *“anomalous refraction refers to more exotic phenomena.”* In the revised version, we have removed this terminology.

-In Fig. 4 caption title and in other parts of the text: “the most general case of refraction” can just read “general case of refraction”. Likewise, in the second to last paragraph, “most extreme anisotropic case” can just be “extreme anisotropic case.” Many instances of using “very” in the main text and SOM can be removed. The authors should avoid other unnecessary superlatives.

We agree with the referee that we should avoid unnecessary superlatives. In the revised version, we have removed “most”, “very” and other unnecessary superlative words.

The term “diffraction-free” lensing appears in the title but is never explained.

We thank the referee for this insightful comment. We realize that the term “diffraction-free lensing” might cause misunderstanding among readers since the definition of “diffraction-free” depends on which polaritonic wavevector we are comparing. In order to avoid such confusion, we have modified the title (highlighted in red) and removed the term “diffraction-free”.

Planar nano-optics in anisotropic media: refraction and lensing of in-plane hyperbolic polaritons

There are some problems with the “bending-free refraction” part of this work:

Throughout the text, the authors give the impression that “bending-free” refraction is only possible in hyperbolic media because of their “unique properties”, only to finally say that it is also possible in non-hyperbolic anisotropic media in the second-to-last paragraph. The explanation of why this occurs seems to rely on the asymptotes of the hyperbola, so it is unclear why this would be a more general phenomenon. Furthermore, the statement “IFCS... becomes two parallel straight lines” does not make sense. The IFCS are not changing, the k -vectors are approaching the IFC asymptotes. The author should provide a better general intuitive understanding of why bending-free refraction occurs.

We thank the referee for this insightful comment and agree that the IFCS are not changing. The IFCS describe two open hyperbolas with similar shapes in the effective media $\text{MoO}_3/\text{SiO}_2$ and MoO_3/air . The referee is correct: the bending-free refraction is not a general phenomenon in all hyperbolic media since its prerequisite would be that the two hyperbolic IFCS have similar shapes. In that case, the tangents to both hyperbolas become parallel when \mathbf{k}_{in} and \mathbf{k}_{out} approach the asymptotes. Since \mathbf{S} generally remains perpendicular to the tangent to the IFC, the refracted wave propagates almost parallel to the incident wave (i.e. $\mathbf{S}_{\text{in}} \parallel \mathbf{S}_{\text{out}}$), as if the incident wave had been directly transmitted without any change in its propagation direction. This is what we term bending-free refraction. In order to make the description of bending-free refraction clearer, we have added the following text (highlighted in red) in the revised version:

“The boundary is also tilted a given angle with respect to the crystal axes. Interestingly, due to the similar shapes of the IFCs in the considered hyperbolic media (α -MoO₃/air and α -MoO₃/SiO₂, black and grey curves in Fig. 1b, respectively), the Poynting vectors of the incident and refracted waves are parallel for almost any \mathbf{k}_{in} and boundary angle φ , especially in the region where the arms of both hyperbolic IFCs are straight. Hence, the refracted wave propagates almost parallel to the incident wave (i.e. $\theta_{\text{in}-S} \approx \theta_{\text{out}-S}$), as if the incident wave had been transmitted directly without any change in its propagation direction (black and blue arrows in Fig. 1d). This feature opens the door to the realization of bending-free refraction at arbitrary incident angles in anisotropic media, which is not possible in isotropic media.”

“...Furthermore, we highlight that, since \mathbf{k}_{in} and \mathbf{k}_{out} are close to the asymptotes of the IFCs, where the tangents to both hyperbolas are parallel, \mathbf{S}_{in} and \mathbf{S}_{out} are almost parallel after refraction at boundary-3. As a result, polaritons refract upon boundary-3 behave as if they had been directly transmitted without any change in their direction of propagation, despite the wavevector does change upon this refraction phenomenon. Therefore, refraction upon this prism is bending-free, in excellent agreement with our theoretical prediction shown in Fig. 1d. Such bending-free refractive prism showed in our work will open exciting possibilities to engineer polaritonic wavefronts at the nanoscale without the need of changing their direction of propagation.”

REVIEWERS' COMMENTS

Reviewer #2 (Remarks to the Author):

The authors have improved their MS. OK to publish.